# Roles of adipose-derived stem cells and derived exosomes in therapeutic applications to testicular injury caused by cisplatin

**Shixuan Wu**[ID]**, Kunlong Lv, Tao Zheng, Tianbiao Zhang, Yonghao Nan, Rui Wang***

Department of Andrology, The First Affiliated Hospital of Zhengzhou University, Zhengzhou, China

* wang998rui@21cn.com

**Data Availability Statement:** I have uploaded the complete data to: figshare:doi:10.6084/m9. figshare.24474409.

## Abstract

In recent years, adipose-derived stem cells (ADSCs) and derived exosomes (ADSC-Ex) have been investigated for their therapeutic potential in various diseases due to their satisfactory differentiation and regeneration ability. We aimed to explore the potential treatment of ADSCs and ADSC-Ex for testicular injury caused by cisplatin. ADSCs and ADSC-Ex s were identified and extracted to treat the rat model with testicular injury caused by cisplatin. Then the immunohistochemistry and Enzyme linked immunosorbent assay (ELISA) were used to detect the potential treatment of ADSCs and ADSC-Ex. We found that ADSCs and ADSC-Ex significantly improved the testicular tissue damage, increased the number of germ cells, and improved the arrangement of the seminiferous tubules. The levels of malondialdehyde and testosterone were also improved. We speculated that ADSCs and ADSC-Ex may alleviate the testicular injury caused by cisplatin.

## 1. Introduction

Cisplatin is a widely used chemotherapy drug that can cause testicular damage, including germ cell loss, Leydig cell dysfunction, and damage to the blood-testis barrier [1]. This can lead to infertility and hormonal imbalances in male cancer survivors [2].

Adipose-derived stem cells (ADSCs) are a type of mesenchymal stem cell that can be isolated from adipose tissue [3]. ADSCs have the ability to differentiate into multiple cell types, including adipocytes, osteoblasts, chondrocytes, and myocytes [4]. In addition, ADSCs secrete various growth factors and cytokines that promote tissue repair and regeneration [5,6]. Exosomes are small membrane-bound vesicles that can transport proteins, nucleic acids, and other molecules between cells [7]. Exosomes derived from ADSCs, known as ADSC-Ex, have been shown to have similar therapeutic effects as ADSCs themselves [8]. ADSC-Ex can promote tissue repair and regeneration, reduce inflammation, and modulate immune responses [9,10].

In recent years, ADSCs and ADSC-Ex have been investigated for their therapeutic potential in various diseases and conditions, including wound healing, tissue regeneration, autoimmune diseases, and neurological disorders [6,11]. In preclinical studies, ADSCs and adipose-derived

**Funding:** Rui Wang was supported by the Henan Provincial Health Care Commission Provincial Ministry Project Fund (LHGJ20190276). The funders had no role in study design, data collection and analysis, decision to publish, or preparation of the manuscript.

**Competing interests:** The authors have declared that no competing interests exist.

mesenchymal stem cell extracellular vesicles (ADSC-EVs) have shown promising results in improving tissue function and reducing inflammation in animal models of these diseases [12].

However, few studies reported the effect of ADSC-Ex treatment on testicular injury. In this study, we aimed to extract the ADSC-Ex and explore the impact of ADSC-Ex on rat models with testicular injury.

## 2.Methods

### 2.1. Model establishment

All rats were maintained group-housed under specific pathogen free (SPF) conditions with a 12h/12h light dark cycle at 23˚c with free access to food and water at the animal facility of the First Affiliated Hospital of Zhengzhou University according to the Institutional Animal Committee Guidelines. Methods of sacrifice: The animals were anesthetized by intraperitoneal injection of sodium pentobarbital and then sacrificed. Efforts to alleviate suffer: Fully consider the interests of animals, treat animals well, prevent or reduce stress, pain and injury to animals, respect animal life, stop barbaric behavior against animals, and use the least painful method to deal with animals. The animal model of testicular cell damage was established by intraperitoneal injection of a 10mg/kg cisplatin solution. Three days after modeling, the testicular cell damage model + cell therapy group received a tail vein injection of $3*10^6$ CM-DiI-labeled ADSCs cells, while the testicular cell damage model + Ex group received a tail vein injection of 100μg of ADSC-Ex The normal control group and the testicular cell damage model group received an equal volume of saline via tail vein injection [13–15]. The procedures of Dil labeling were as follow: remove the supernatant from the culture dish and replace it with serum-free basal medium dulbecco's modified eagle medium/nutrient mixture F-12 (DMEM/F12) for cell cultivation. After 48 hours of incubation, collect the supernatant for extracellular vesicle extraction by the molecular department. For CM-DIL labeling of cells, digest the cells and suspend them in 10% DF/12 medium containing 5ul/ml of CM-DIL. Incubate the cell suspension at 37˚C for 5 minutes and then place it at 4˚C for 15 minutes to meet the experimental requirements. Provide a cell suspension with a concentration of $3\times10^6$ cells/ml labeled with CM-DIL for injection into the animal facility. Serum and bilateral testes were collected for pathological examination 24 hours after extracellular vesicle injection.

### 2.2. Cell identification

Six-week-old Sparaque Dawley rat was disinfected with 75% alcohol for 3 minutes. The rat was then placed on an ice plate in a laminar flow hood, and subcutaneous white adipose tissue was quickly removed from the inguinal area of the rat [16]. Visible blood vessels and fibrous parts were removed from the adipose tissue, which was then washed three times with PBS, cut into small pieces, and digested with 0.2% collagenase I at 4˚C overnight. The next day, the tissue was further digested at 37˚C for 20 minutes with shaking every 5 minutes. The digestion was stopped by adding a complete containing 10% fetal bovine serum (FBS) and 1% penicillin–streptomycin (P/S). The culture medium, FBS, P/S were purchased from Gibco (Thermo-Fisher, Shanghai, China). The tissue digestion mixture was filtered through a 70-μm cell strainer, and the filtrate was collected and centrifuged at 1000 rpm for 5 minutes. The supernatant was discarded, and the cell pellet was resuspended in a fresh culture medium and cultured in a culture flask. The third passage of cells was collected by removing the culture medium and washing the cells twice with 1×PBS. The cells were then digested with 0.25% trypsin (containing 0.02% ethylenediaminetetraacetic acid (EDTA)) until they became round. The digestion was stopped by adding a culture medium, and the cell suspension was collected and

centrifuged at 1000 rpm for 3 minutes. The supernatant was discarded, and the cell pellet was resuspended in PBS and centrifuged again before being analyzed by flow cytometry [13].

## 2.3. Exosome extraction

Thaw the cell sample at 37˚C. Transfer the sample to a new centrifuge tube and centrifuge at $2000 \times g$, 4˚C for 30 minutes. Carefully transfer the supernatant to a new centrifuge tube and centrifuge again at $10,000 \times g$, 4˚C for 45 minutes to remove larger vesicles [14]. Collect the supernatant, filter it through a 0.45 μm membrane, and collect the filtrate. Transfer the filtrate to a new centrifuge tube and centrifuge at $100,000 \times g$, 4˚C for 70 minutes using an ultracentrifuge rotor. Discard the supernatant, resuspend the pellet in 10 mL of pre-cooled 1×PBS, and centrifuge again at $100,000 \times g$, 4˚C for 70 minutes using an ultracentrifuge rotor. Discard the supernatant, resuspend the pellet in 100 μL of pre-cooled 1×PBS, and take 20 μL for electron microscopy, 10 μL for particle size analysis, and 20 μL for fluorescence. The remaining extracellular vesicles are stored at -80˚C.

## 2.4. Flow cytometry

Cells were collected, centrifuged to remove the supernatant, and washed once with 1 mL of PBS at 1500 rpm for 5 min. The cells were divided into six tubes, one tube being the negative control, and the other five tubes were added with 5 μL of CD29 PE, CD45 PERCP/CY5.5, CD90 FITC, CD34 PE, and CD11b/c FITC antibodies, respectively. After gentle mixing, the tubes were incubated at room temperature in the dark for 20 min. Then, 1 mL of PBS was added, the tubes were centrifuged at 1500 rpm for 5 min, and the supernatant was discarded. This step was repeated once. The cells were resuspended in 500 μL of PBS and analyzed by flow cytometry.

## 2.5. Evaluation of ADCS-Ex

Transmission electron microscopy observation of extracellular vesicle samples: 10 μL of extracellular vesicles were taken and dropped onto a copper grid for 1 min. The excess liquid was removed with filter paper. 10 μL of uranyl acetate was added to the copper grid for 1 min, and the excess liquid was removed with filter paper. The copper grid was air-dried for several minutes and observed by transmission electron microscopy to obtain the imaging results. Fluorescent labeling and nano flow cytometry detection of extracellular vesicle samples: 20 μL of extracellular vesicles were diluted to 60 μL, and 30 μL of the diluted extracellular vesicles were added to 20 μL of fluorescently labeled antibodies (CD9, CD63), mixed, and incubated at 37˚C in the dark for 30 min. 1 mL of pre-cooled PBS was added, and the tubes were centrifuged at $110,000 \times g$ at 4˚C for 70 min. The supernatant was carefully removed, and 1 mL of pre-cooled PBS was added. The tubes were centrifuged again at $110,000 \times g$ at 4˚C for 70 min. The supernatant was carefully removed, and the pellets were resuspended in 50 μL of pre-cooled 1×PBS. The instrument performance was tested with standard samples before extracellular vesicle samples were analyzed. Gradient dilution was performed to avoid sample blockage in the sample needle. The protein index results were obtained by detecting the samples using the NanoFCM instrument.

## 2.6. Immunohistochemistry

Tissue sections were baked in a 65˚C oven for 2 hours. The sections were deparaffinized with xylene for 10 minutes, followed by another 10 minutes in fresh xylene. The sections were then sequentially placed in 100% ethanol, 95% ethanol, 80% ethanol, and purified water for 5

minutes each. The sections were placed in a repair box and antigen retrieval was performed by adding antigen retrieval solution (citrate buffer) and heating in a pressure cooker until automatic release of pressure. After 2 minutes, the box was removed from the heat source and allowed to cool naturally. The antigen retrieval solution was discarded, and the sections were washed with PBS. The sections were then treated with 0.5% Triton X-100 (PBS solution) at room temperature for 20 minutes. The sections were washed with PBS for 3 times, each for 5 minutes, and the surrounding PBS was removed by blotting with absorbent paper. 5% bovine albumin (BSA) was added to the sections and incubated at 37°C for 30 minutes to block non-specific binding. The excess blocking solution around the tissue was removed by blotting with absorbent paper, and the sections were not washed. A diluted primary antibody (c-Kit, 1:200) was added to each slide, and the slides were placed in a humid box and incubated at 4°C. The wet box incubated at 4°C overnight was taken out and left at room temperature for 45 minutes. The slides were washed with PBS for 3 times, each for 5 minutes, and then incubated with horseradish peroxidase-labeled goat anti-rabbit IgG (H+L) (1:100) at 37°C for 30 minutes. The slides were washed with PBS thoroughly. Diaminobenzidine (DAB) was used for color development for 5–10 minutes, and the staining intensity was observed under a microscope. The slides were washed with PBS or tap water for 1 minute, followed by counterstaining with hematoxylin for 3 minutes and differentiation with hydrochloric acid alcohol until blue. The slides were washed with tap water for 1 minute, dehydrated, mounted, and examined under a microscope.

## 2.7. Histopathological evaluation

Testicular tissues were obtained and subjected to dehydration with 70%, 80%, and 90% ethanol solutions, followed by immersion in a mixture of equal volume of pure alcohol and xylene for 15 minutes, xylene I for 15 minutes, xylene II until it is transparent. The tissues were then placed in a mixture of half xylene and half paraffin wax for 15 minutes, followed by immersion in paraffin I and paraffin II until transparent for 50–60 minutes each. The tissues were embedded in paraffin and sectioned. The paraffin sections were subjected to baking, deparaffinization, and hydration. The slices that have been distilled in distilled water were then stained with hematoxylin in a solution of distilled water for 3 minutes, differentiated in hydrochloric acid ethanol differentiation solution for 15 seconds, rinsed with tap water, put in a blueing solution for 15 seconds, rinsed with tap water, stained with eosin for 3 minutes, rinsed with water, dehydrated, cleared, sealed, and observed under a microscope. ImageJ software was used to measure the thickness of the epithelium of the seminiferous tubules and the diameter of the seminiferous tubules. To evaluate the degree of testicular damage and spermatogenic ability, the Mean Testicular Biopsy Score (MTBS) (Johnsen, 1970) was used. The score for each small tube was 1–10, based on the germ cell capacity of different tissues (Table 1). A microscope magnified to 400 times was used to evaluate 20 seminiferous tubules in each group to assess the thickness of the epithelium of the seminiferous tubules and the diameter of the seminiferous tubules.

## 2.8. ELISA detection

ELISA kits were used to identify the levels of follicle-stimulating hormone (FSH, E-EL-RO391c, abscience) and luteinizing hormone (LH, E-EL-RO026c, Elabscience), testosterone (T, E-OSEL-R0003, ascience), malondialdehyde (MDA, E-EL-0060c,Elabscience). Standard wells, blank wells, and sample wells were set up. The standard wells were added with 100μL of standard product diluted by a certain ratio, the blank wells were added with 100μL of a standard product and sample dilution, and the remaining wells were added with 100μL of the test sample. The enzyme-labeled plate was covered and incubated at 37°C for 90 minutes.

**Table 1. Testicular biopsy scores (MTBS).**

| Score | Description |
|---|---|
| 1 | No cells |
| 2 | Sertoli cells without germ cells |
| 3 | Only spermatogonia |
| 4 | Only a few spermatocytes |
| 5 | Many spermatocytes |
| 6 | Only a few early spermatids |
| 7 | Many early spermatids without differentiation |
| 8 | Few late spermatids |
| 9 | Many late spermatids |
| 10 | Full spermatogenesis |

The liquid in the wells was discarded without washing. Each well was added with 100μL of biotinylated antibody working solution, and the enzyme-labeled plate was covered and incubated at 37˚C for 1 hour. The liquid in the wells was discarded, and the plate was blotted dry on clean absorbent paper. Each well was added with 350μL of washing solution, soaked for 1 minute, and the liquid in the enzyme-labeled plate was sucked or shaken off and blotted dry. This washing step was repeated 3 times. After washing, the plate should be used immediately for the next step and should not be allowed to dry. Each well was added with 100μL of enzyme conjugate working solution, and the enzyme-labeled plate was covered and incubated at 37˚C for 30 minutes. The liquid in the wells was discarded, and the plate was washed 5 times, following the same method as in step 3. Each well was added with 90 μL of substrate solution (Tetramethylbenzidine (TMB)), and the enzyme-labeled plate was covered and incubated at 37˚C for about 15 minutes in the dark. The enzyme-labeled instrument was preheated 15 minutes in advance. Each well was added with 50 μL of stop solution to terminate the reaction. Note: The order of adding the stop solution should be as consistent as possible with that of the substrate solution. The optical density (OD value) of each well was immediately measured at 450 nm wavelength using an enzyme-labeled instrument.

## 2.9. Statistical analysis

All data were analyzed using SPSS 22.0 and presented as mean ± SD. One-way ANOVA was used to analyze the significant differences between groups, with $P<0.05$ as the criterion for statistical significance. Benjamini and Hochberg method was used to adjusted the p-values.

## 2.10. Ethical statement

The authors are accountable for all aspects of the work in ensuring that questions related to the accuracy or integrity of any part of the work are appropriately investigated and resolved. The study was conducted in accordance with the Declaration of Helsinki (as revised in 2013). Experiments were performed under a project license(No.ZZU-LAC20190125) granted by the Animal Ethical and Welfare Committee, in compliance with the ethics committee of the First Affiliated Hospital of Zhengzhou University.

## 3.Results

### 3.1. Models validation

The results, as shown in Fig 1, revealed that the pathological results (Fig 1A) showed a significant decrease in the layer of germ cells and a sparse arrangement of the seminiferous tubules

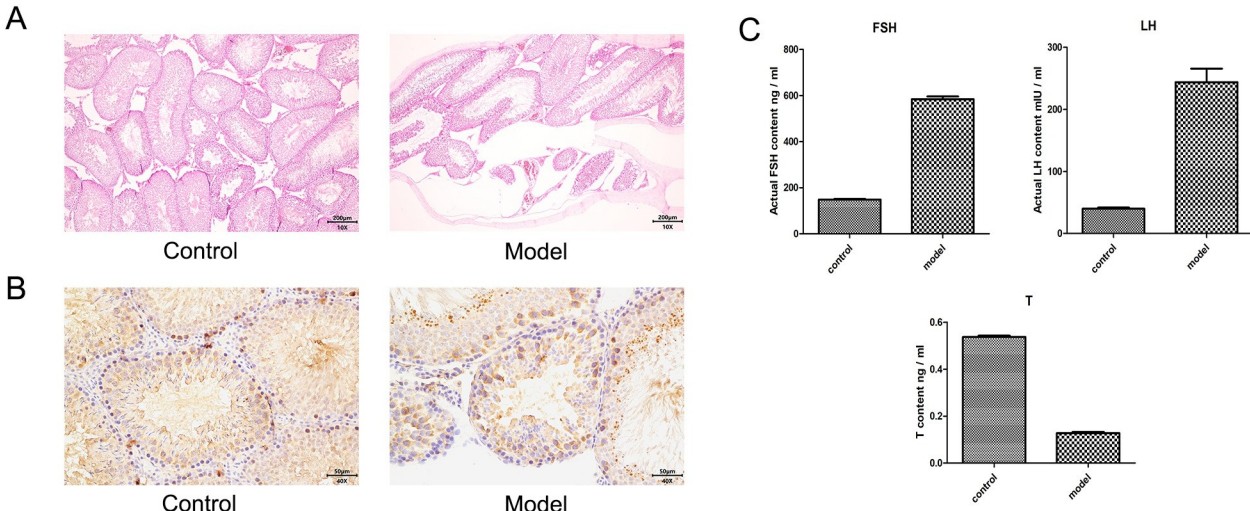

**Fig 1. Results of models validation.** A: Showed a significant decrease in the layer of germ cells and a sparse arrangement of the seminiferous tubules in the testicular tissue of the model group compared to the control group. B: Showed a significant decrease in the expression level of c-Kit in the model group compared to the control group. C: Showed a significant increase in the levels of follicle-stimulating hormone (FSH) and luteinizing hormone (LH), a significant decrease in the level of testosterone (T).

in the testicular tissue of the model group compared to the control group. The immunohisto-chemical results (Fig 1B) showed a significant decrease in the expression level of c-Kit in the model group compared to the control group. The ELISA results (Fig 1C) showed a significant increase in the levels of follicle-stimulating hormone (FSH) and luteinizing hormone (LH), a significant decrease in the level of testosterone (T), and the differences were statistically significant (P<0.05).

## 3.2. Identification of ADSCs cells

The results, as shown in Fig 2, revealed that the flow cytometry analysis showed positive expression of CD29 and CD90 and negative expression of CD11b, CD34, and CD45 in the cells (Fig 2A). The Alizarin Red staining indicated that the cells had the ability of osteogenic differentiation, suggesting the successful isolation of adipose-derived stem cells (Fig 2B).

## 3.3. Evaluation of ADSC-Ex

Identification of ADSC-Ex through transmission electron microscope (TEM), Nanoparticle Tracking Analysis (NTA), and nanoflow cytometry analysis is shown in Fig 3. The morphology and size of ADSC-Ex were evaluated using TEM (Fig 3A) and NTA (Fig 3B). The electron microscopy and particle size results showed that the ADSC-Ex had a double-layered membrane of approximately 80nm in diameter, with a cup-shaped structure. The nanoflow cytometry results showed that the positivity rate of the ADSC-Ex labeled with CD63 and CD9 was higher than that of the blank control group (Fig 3C).

## 3.4. Effect of ADSC-Ex on testicular injury models

The results shown in Fig 4 revealed that the pathological results (Fig 4A) showed a significant decrease in the layer of germ cells and a sparse arrangement of the seminiferous tubules in the testicular tissue of the model group compared to the control group. The ADSCs cells and their ADSC-Ex significantly improved the testicular tissue damage, increased the number of germ

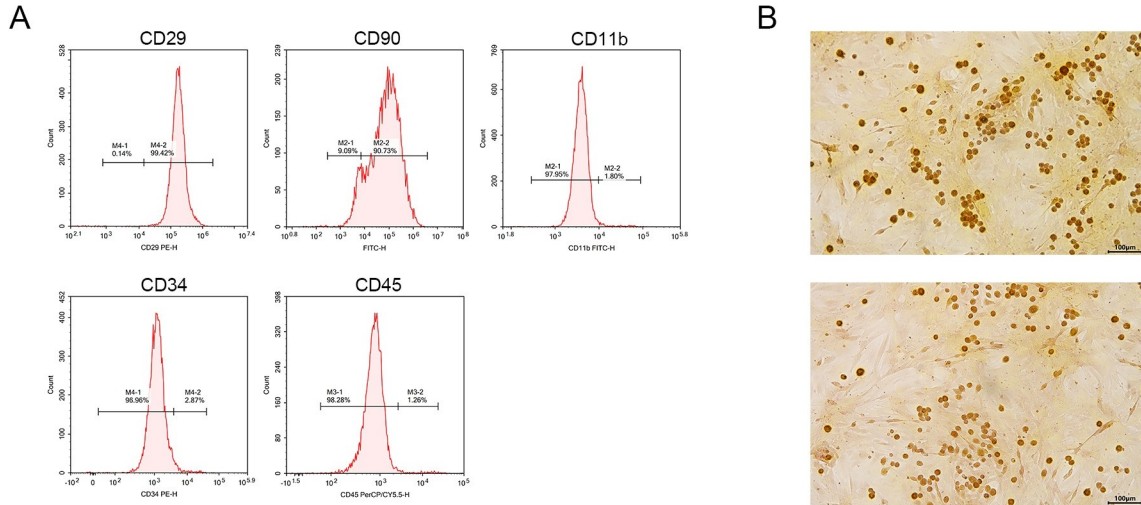

**Fig 2.** Results of ADSC-Ex: A: Showed positive expression of CD29 and CD90 and negative expression of CD11b, CD34, and CD45 in the cells; B: Results of Alizarin Red staining.

cells, and improved the arrangement of the seminiferous tubules (Fig 4B–4D). The immuno-histochemical results (Fig 5A and 5B) showed a significant decrease in the expression level of c-Kit in the model group compared to the control group, while the expression level of c-Kit in the testicular tissue was significantly increased after the intervention of ADSCs cells and their ADSC-Ex. The ELISA results (Fig 5C) showed a significant increase in the levels of malondial-dehyde (MDA) in the model group compared to the control group, and a significant decrease in the levels of MDA in the ADSCs cells and their ADSC-Ex treated groups compared to the model group, with statistical significance (P<0.05). Similarly, testosterone (T) levels were

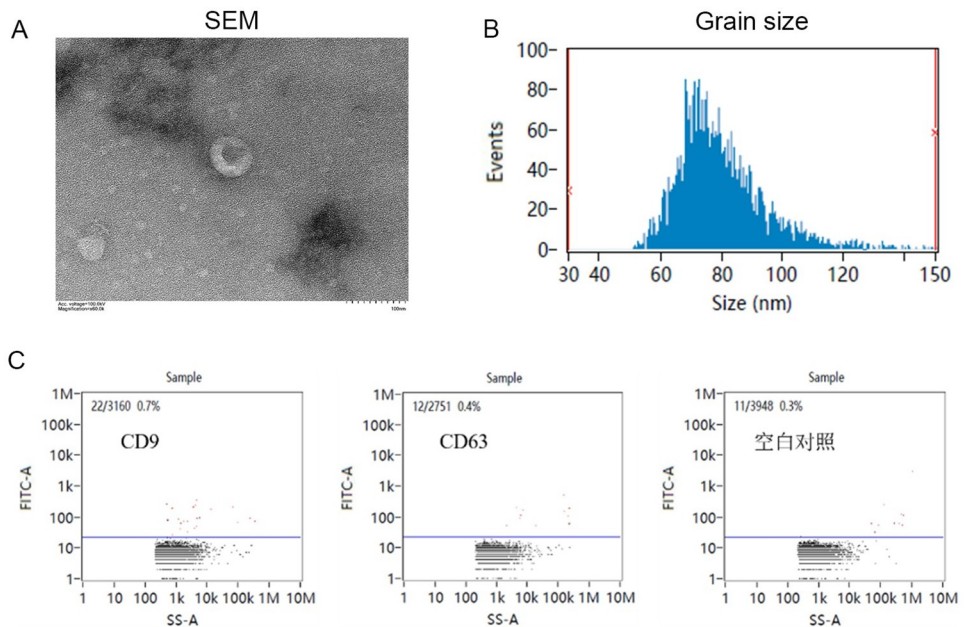

**Fig 3. Results of evaluation of ADSC-Ex.** A-B: Morphology and size of ADSC-Ex were evaluated using TEM and NTA. C: Positivity rate of the ADSC-Ex labeled with CD63 and CD9 was higher than that of the blank control group.

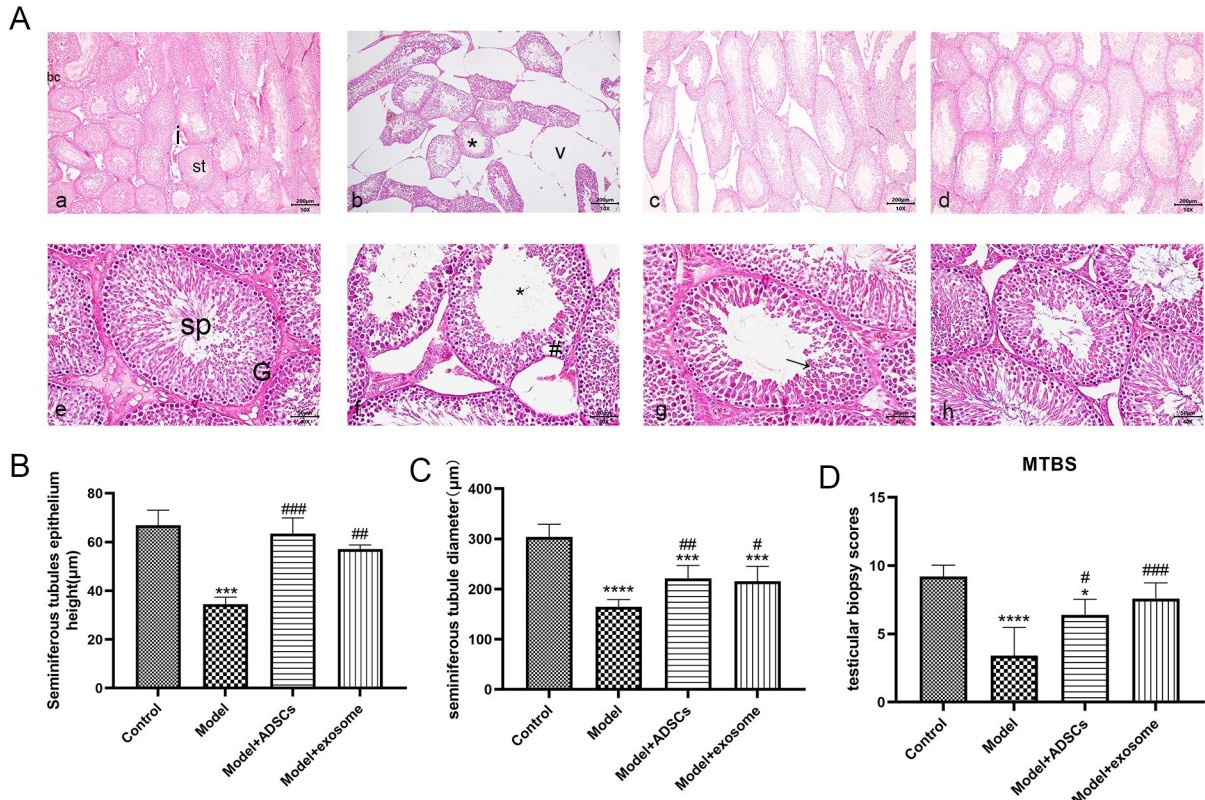

**Fig 4. Histopathological evaluation of the effect of ADSC-Ex on testicular injury models.** A: Results of histopathology, H&E stained sections in the testis of rat, (a,c) for (G I), (b,f) for (G II), (c,g) for (G III), (d,h) for (G IV), showing: **a** seminiferous tubules (st), interstitial tissue (i) and blood capillary (bc), **b** large interstitial vacuoles (v), distorted and reduced seminiferous tubules(*), **e** spermatogonia (G), sperm (sp), **f** Irregular and deformed seminiferous tubule epithelium(*),sloughing of degenerated spermatogenic cells(#), **g** restoration of epithelium thickness (arrow), **h** Restoration of the normal architecture of seminiferous tubule.B-D: Changes of seminiferous tubules epithelium, diameter and testicular biopsy scores.

significantly decreased in the model group compared to the control group, and both the treated groups rescued the decrease of T (Fig 5D).

## 4. Discussion

In this study, we aimed to use adipose-derived mesenchymal stem cell-derived exosomes to treat rats with cisplatin-induced testicular injury. By using TEM, NTA, and nanoflow cytometry analysis, we successfully isolated and identified the ADSC-Ex. The results showed that the ADSC-Ex had a double-layered membrane of approximately 80nm in diameter, with a cup-shaped structure, and were positive for CD63 and CD9. The in vivo experiments demonstrated that the administration of ADSC-Ex significantly improved the testicular tissue damage, increased the number of germ cells, and improved the arrangement of the seminiferous tubules. The results of the immunohistochemical and ELISA analyses further supported the therapeutic effect of the ADSC-Ex.

Cisplatin-induced testicular injury is a well-known side effect of chemotherapy, which can lead to infertility and other reproductive problems [17]. The mechanism of cisplatin-induced testicular injury is complex and involves oxidative stress, inflammation, and DNA damage [18]. Cisplatin can induce the production of reactive oxygen species (ROS) and reduce the activity of antioxidant enzymes, leading to oxidative stress and damage to testicular cells [1].

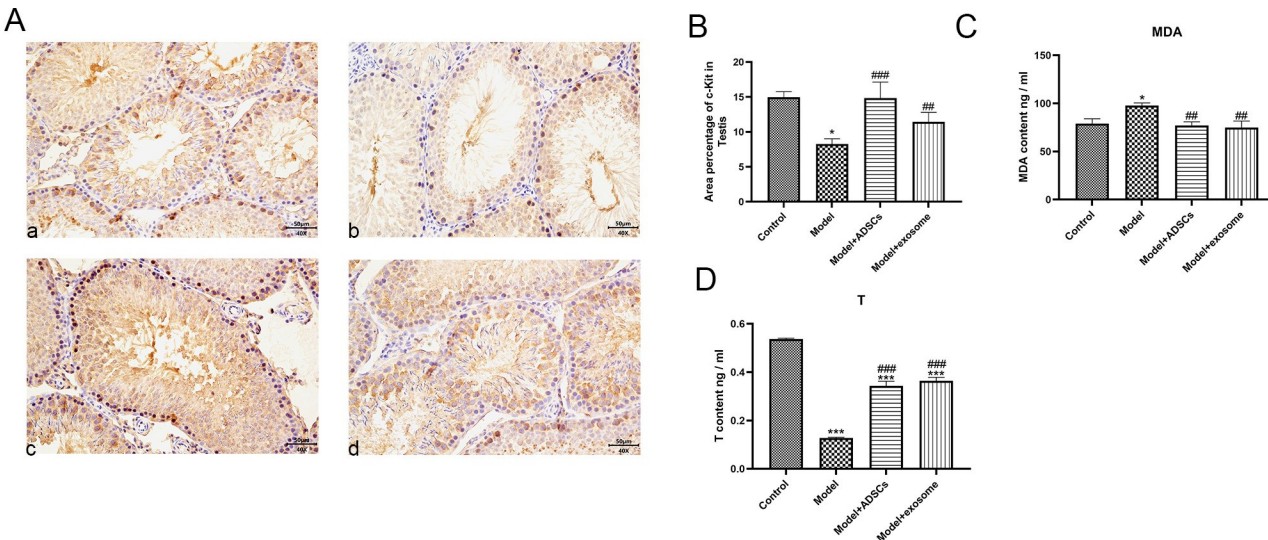

**Fig 5. Effect of ADSC-Ex on testicular injury models.** A-B: Immunohistochemical results. C-D: Changes of MDA content and T content.

In addition, cisplatin can activate the Nuclear factor kappa-B (NF-κB) signaling pathway and increase the expression of pro-inflammatory cytokines, leading to inflammation and tissue damage [19]. Furthermore, cisplatin can directly damage DNA and induce apoptosis in testicular cells [15].

Clinical studies have shown that cisplatin-induced testicular injury is common in cancer patients receiving chemotherapy, and can lead to a significant reduction in sperm count and motility [20]. In addition, cisplatin-induced testicular injury can also lead to hormonal imbalances and other reproductive problems. Various strategies have been proposed to prevent or reduce cisplatin-induced testicular injury, including the use of antioxidants, hormone replacement therapy, and cryopreservation of sperm prior to chemotherapy [1,21].

Basic research on cisplatin-induced testicular injury has focused on understanding the underlying mechanisms and developing new therapeutic strategies. Animal models have been used to study the effects of cisplatin on testicular function, and to evaluate the efficacy of potential treatments [22]. Recent studies have suggested that compounds such as melatonin, resveratrol, and curcumin may have protective effects against cisplatin-induced testicular injury by reducing oxidative stress and inflammation [23].

ADSC-Ex are small membrane-bound particles secreted by various cell types, including adipose-derived mesenchymal stem cells (ADSCs). ADSC-Ex plays a crucial role in intercellular communication, as they can transfer bioactive molecules, such as proteins, lipids, and nucleic acids, to recipient cells [24]. In recent years, there has been a growing interest in the therapeutic potential of ADSC-Ex in various diseases, including tissue injury, inflammation, and cancer [25].

ADSC-Ex has several advantages over ADSCs themselves, including lower immunogenicity, reduced risk of tumorigenesis, and easier storage and transportation [26]. Moreover, ADSC-Ex can cross biological barriers, such as the blood-brain barrier, and target specific tissues or cells, making them an attractive therapeutic option [27] Several studies have investigated the therapeutic potential of ADSC-Ex in tissue injury models, such as myocardial infarction, acute kidney injury, and spinal cord injury [28]. These studies have shown that ADSC-Ex can promote tissue repair, reduce inflammation, and improve functional recovery [29].

In addition to tissue injury, ADSC-Ex has also been studied in cancer therapy. ADSC-Ex can deliver anti-tumor agents, such as miRNAs and chemotherapeutic drugs, to cancer cells and inhibit tumor growth [30]. Moreover, ADSC-Ex can modulate the immune response and enhance the anti-tumor activity of immune cells [31].

In our study, we isolated ADSCs from inguinal adipose tissue of rats. As previously reported [32], we also used ultracentrifugation-based isolation technique to isolate double-layered membrane microvesicles of about 80nm in size, similar to "cup-shaped" exosomes, which is consistent with previous studies.

Chemotherapeutic drugs cannot distinguish between cancer cells and normal cells, thus causing damage to normal organs [33]. Especially after damaging the testis, it will have adverse effects on testicular tissue, blood, and semen parameters. This study confirms this view. At the same time, we also confirmed the damaging effect of cisplatin, a chemotherapeutic drug widely used in urological tumors (especially testicular cancer), on testicular tissue and germ cells, which is consistent with previous research conclusions [34,35].

Regarding the evaluation under an optical microscope, we found that testicular tissue exposed to cisplatin had obvious damage, manifested as a decrease in the thickness of the epithelium of the seminiferous tubules, a decrease in the diameter of the seminiferous tubules, and a large number of vacuoles in the interstitial cells. These changes caused a decrease in rat testicular spermatogenic function, similar to rats exposed to cisplatin and rats undergoing ischemia-reperfusion [36,37].

C-kit, as a stem cell factor receptor and a marker for distinguishing spermatogenic cell types, is strongly positive in all germ cells except for sperm cells. Our immunohistochemical results showed that the number of c-kit positive cells and the expression of all germ cells in the cisplatin exposure group were significantly decreased, which is consistent with previous research findings. However, treatment with ADSCs and exosomes significantly increased the number of c-kit positive cells [38].

The ROS induced by cisplatin can induce oxidative stress, causing damage to testicular tissue [17]. This damage can lead to an imbalance between pro-oxidants and antioxidants. Previous studies have found that MDA, as an oxidative stress marker, is produced by the peroxidation of lipids by ROS [1]. It affects the activity of mitochondrial respiratory chain complexes and key enzymes in the mitochondria in vitro. In this study, the increase of MDA in the cisplatin exposure group confirmed this finding [39]. The protective effect of ADSCs and their exosomes against oxidative stress can be attributed to the antioxidant capacity of stem cells through ROS clearance and the direct delivery of antioxidant enzyme mRNA or proteins by exosomes, thus protecting testicular tissue from oxidative stress, apoptosis, and promoting tissue regeneration.

Despite the promising results of our study, there are several limitations that should be considered. Firstly, the current study was performed in a rat model of cisplatin-induced testicular injury, and further studies are needed to confirm the therapeutic effects of adipose-derived stem cells (ADSCs) and their exosomes in humans. Secondly, the mechanisms underlying the therapeutic effects of ADSCs and their exosomes on testicular injury are still not fully understood, and more research is needed to elucidate these mechanisms. Thirdly, the long-term safety and potential side effects of ADSCs and their exosomes need to be carefully evaluated in future studies. Finally, the cost and practicality of using ADSCs and their exosomes as a therapeutic approach for testicular injury in clinical settings need to be further explored. Moreover, several challenges need to be addressed before ADSC-Ex can be used in clinical setting [40]. These include standardization of isolation and characterization methods, optimization of ADSC-Ex production and storage, and identification of the optimal dosing and delivery

routes. Nevertheless, ADSC-Ex represents a promising therapeutic option with great potential for clinical translation.

## 5.Conclusion

In this study, the ADSC-Ex were extracted and identified, and the effectiveness of adipose-derived stem cells and derived exosomes on testicular injury caused by cisplatin were proved.

## Author Contributions

**Data curation:** Kunlong Lv, Tao Zheng, Tianbiao Zhang, Yonghao Nan.

**Formal analysis:** Tao Zheng.

**Resources:** Shixuan Wu, Yonghao Nan.

**Software:** Shixuan Wu, Kunlong Lv.

**Supervision:** Shixuan Wu, Rui Wang.

**Validation:** Shixuan Wu.

**Visualization:** Shixuan Wu.

**Writing – original draft:** Shixuan Wu.

**Writing – review & editing:** Shixuan Wu.

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
