## [Decision Letter · Decision Letter 0]

11 Oct 2023

PONE-D-23-25567Adipose-derived stem cells and derived exosomes may alleviate the testicular injury caused by cisplatin.PLOS ONE

Dear Dr. wu,

Thank you for submitting your manuscript to PLOS ONE. After careful consideration, we feel that it has merit but does not fully meet PLOS ONE’s publication criteria as it currently stands. Therefore, we invite you to submit a revised version of the manuscript that addresses the points raised during the review process. Comments from PLOS Editorial Office: We note that one or more reviewers has recommended that you cite specific previously published works. As always, we recommend that you please review and evaluate the requested works to determine whether they are relevant and should be cited. It is not a requirement to cite these works. We appreciate your attention to this request.

We look forward to receiving your revised manuscript.

Kind regards,

Ahmed E. Abdel Moneim

Academic Editor

PLOS ONE

Journal Requirements:

"Rui Wang was supported by the Henan Provincial Health Care Commission Provincial Ministry Project Fund (LHGJ20190276)."

7. Please amend the manuscript submission data (via Edit Submission) to include author Dr. Kunlong Lv, Dr. Tao Zheng, Dr. Tianbiao Zhang, Dr. Yonghao Nan and Dr. Rui Wang. 

8. Please amend your manuscript to include your abstract after the title page.

9. Your ethics statement should only appear in the Methods section of your manuscript. If your ethics statement is written in any section besides the Methods, please move it to the Methods section and delete it from any other section. Please ensure that your ethics statement is included in your manuscript, as the ethics statement entered into the online submission form will not be published alongside your manuscript. 

Reviewers' comments:

Reviewer's Responses to Questions

**Comments to the Author**

1. Is the manuscript technically sound, and do the data support the conclusions?

Reviewer #1: Yes

Reviewer #2: Partly

Reviewer #3: Partly

Reviewer #4: No

2. Has the statistical analysis been performed appropriately and rigorously? 

Reviewer #1: Yes

Reviewer #2: Yes

Reviewer #3: No

Reviewer #4: No

3. Have the authors made all data underlying the findings in their manuscript fully available?

Reviewer #1: Yes

Reviewer #2: Yes

Reviewer #3: Yes

Reviewer #4: Yes

4. Is the manuscript presented in an intelligible fashion and written in standard English?

Reviewer #1: Yes

Reviewer #2: No

Reviewer #3: Yes

Reviewer #4: Yes

5. Review Comments to the Author

Reviewer #1: The manuscript entitled “Adipose-derived stem cells and derived exosomes may alleviate the testicular injury caused by cisplatin" appears to be interesting, but there are many flaws and concerns on it. Study can be greatly improved if following suggestions were incorporated.

1. The title of the paper is not accurately expressed, and I think it needs to be rewritten.

2. Abstract should have Background information on role adipose-derived stem cells and derived exosomes in testicular injury

3. Paper is replete with spelling mistakes. Please, edit the manuscript regarding word mistakes.

4. Some references missing.

The following reference may increase the reader’s comprehension:

a. The following references may increase the reader's comprehension:

Sheykhhasan M, Amini R, Soleimani Asl S, Saidijam M, Hashemi SM, Najafi R. Neuroprotective effects of coenzyme Q10-loaded exosomes obtained from adipose-derived stem cells in a rat model of Alzheimer's disease. Biomed Pharmacother. 2022 Aug;152:113224. doi: 10.1016/j.biopha.2022.113224. Epub 2022 Jun 6. PMID: 35679720.

b. Sheykhhasan M, Heidari F, Eslami Farsani M, Azimzadeh M, Kalhor N, Ababzadeh S, Seyedebrahimi R. Dual role of Exosome in neurodegenerative diseases: a review study. Curr Stem Cell Res Ther. 2023 Jul 26. doi: 10.2174/1574888X18666230726161035. Epub ahead of print. PMID: 37496136.

5. The materials and Methods section requires more information: To confirm the success of ADSC isolation, in addition to the Flow cytometric analysis, it is necessary to confirm using the analysis of the trilineage differentiation potential of ADSC, including histochemical assays (oil red, alkaline phosphatase, and alcian blue stains). Why, there were no other staining for lipid, and chondrocytes? As a result, it is better to include the results of histochemical assays (alkaline phosphatase, and alcian blue stains) in the method and result sections of the present paper.

6. It is recommended that the authors describe the protocol for DiG-labeled ADSCs cells in method section.

7. In "Methods" section, why is the number of cells used in the cell therapy group 3x106, but in the Ex group, 100μg is used. Can the use of two different criteria be scientifically justified?

8. Figure 3 needs correction

9. In order to make the paper more interesting to read, I suggested that the authors could add one graphical abstract to the manuscript.

Reviewer #2: The manuscript entitled Adipose-derived stem cells and derived exosomes may alleviate the testicular injury caused by cisplatin by Wu et al explore the potential treatment of adipose-derived stem cells and derived exosomes for testicular injury caused by cisplatin. The topic covered by the article is interesting as it directs the research on possible alternatives on the control of the testicular damage caused by anticancer drugs (cisplatin).

The presentation of the paper requires more description and clarification of some parts of the documents. Therefore, I recommend that a major revision is warranted, and it should follow the rules of scientific writing and special attention for the references which were not mentioned in methodology section. I explain my concerns below. I ask that the authors specifically address each of my comments in their response.

Comments:

1- kindly follow the rules of scientific writing and special attention for the references

2- kindly revise the manuscript for English errors and grammar

3- Introduction: First page, last line (No. 21); please correct ADSC-EVs to (ADSC-EXs)

4- Methods:

2.1. Model establishment:

The authors did not mention the environment of animal housing, number and model of animals used in experiment.

There are no references for the dose and rout of cisplatine and/or ADSC and/or ADSC-Ex administration.

There are grammatical and spelling errors.

How can you confirm testicular damage after 3 days of cisplatine administration?

2.2. Cell identification

How many rats were used for collecting adipose tissue?

The authors did not mention any references for collection of ADSC form rats, is it a new technique established by the authors?

How can you confirm the viability and count of ADSC after collection?

2.3. Exosome extraction: Again, the authors did not mention any references for collection of exosomes. The authors mentioned (Thaw the sample at 37°C.), but he did not mention which sample and what is the type of sample.

2.4. Flow Cytometry: Kindly please check the soundness of procedures and mention your guide for these procedures.

2.5. Evaluation of ADCS-Ex: No references mentioned for the technique of ADCS-Ex evaluation.

2.6. Immunohistochemistry: What is the method of tissue preservation for immunohistochemistry?

The word (100% ethanol) is repeated twice, please check and correct with referral to reference of this technique. Please mention the source of chemicals and kits used in this study.

2.8. ELISA detection: kindly check references and grammatical errors. The authors did not declare which type of ELISA used and what they searched for?

5- Statistical analysis: What is the post hoc. test was used in your study?

6- Discussion: There is a similar study and will be helpful in your manuscript and you did not mention it in your study: (Cisplatin-induced azoospermia and testicular damage ameliorated by adipose-derived mesenchymal stem cells) Biol Res. 2023 Jan 19;56(1):2. doi: 10.1186/s40659-022-00410-5.

Reviewer #3: Major revision requirieds. Major revision requirieds. Major revision requirieds. Major revision requirieds. Major revision requirieds. Major revision requirieds. Major revision requirieds. Major revision requirieds. Major revision requirieds. Major revision requirieds. Major revision requirieds. Major revision requirieds.

Reviewer #4: The manuscript was carefully reviewed

Some of the flaws in the manuscript include the following

What is the difference between this experiment and previous one (Like the article below)? Mention in discussion.

Meligy FY, Abo Elgheed AT, Alghareeb SM. Therapeutic effect of adipose-derived mesenchymal stem cells on Cisplatin induced testicular damage in adult male albino rat. Ultrastruct Pathol. 2019;43(1):28-55. doi: 10.1080/01913123.2019.1572256. Epub 2019 Feb 9. PMID: 30741078.

The last paragraph of introduction the author writes that “However, few studies reported the effect of ADSC-Ex treatment on testicular injury. In this study, we aimed to extract the ADSC-Ex and explore the impact of ADSC-Ex on rat models with testicular injury.” Ok. Please refer to those articles and what has not been done before that you focused on in this study.

Provide a reference for the testicular destruction model by cisplatin solution. Also, what is the volume of injection of cisplatin and ADSCs cells? Also, on what basis were these concentrations of 3*10^6 CM-DiI-labeled ADSCs cells and 100μg of ADSC-Ex used for injection? What is CM-DiI?

What sedation, anesthesia or anesthesia protocol was used to remove subcutaneous white adipose tissue?

What culture mediums were used? Mention the names of the companies related to the preparation of materials? Why didn't you use penicillin/streptomycin?

The order of method steps in the method is not clear. First Model establishmen then Cell identification and evaluation and then Immunohistochemistry and Immunohistochemistry?!!!! ELISA detection for which protein? From which company? Also, the order of the result should be based on the order of the steps of the work method.

What is NTA? When first expressed, it should not be in the form of abbereviation.

In order to confirm the stem cells, Trilineage Differentiation should be done, but in this study, only differentiation to osteogenesis was done.

What was the difference between the results of using ADSCs cells and of ADSC-Ex? Which one is better?

You don’t use Post hoc to analyze the results of your experimental data?

6. PLOS authors have the option to publish the peer review history of their article (what does this mean?). If published, this will include your full peer review and any attached files.

Reviewer #1: **Yes: **Mohsen Sheykhhasan

Reviewer #2: No

Reviewer #3: No

Reviewer #4: No

---

## [Author Response · Author response to Decision Letter 0]

6 Nov 2023

Response to Reviewer 1

1. The title of the paper is not accurately expressed, and I think it needs to be rewritten.

Response:

Thank you for your comments. The title has been changed as “Roles of adipose-derived stem cells and derived exosomes in therapeutic applications to testicular injury caused by cisplatin.” in our revised manuscript.

2. Abstract should have Background information on role adipose-derived stem cells and derived exosomes in testicular injury.

Response:

Thank you for your comments. Relative background information has been added to abstract in our revised manuscript.

3. Paper is replete with spelling mistakes. Please, edit the manuscript regarding word mistakes.

Response:

Thank you for your valuable comments. The whole manuscript has been revised with the guidance of native English speaker, and all the spelling mistakes have been corrected. 

4. Some references missing.

The following reference may increase the reader’s comprehension:

a. The following references may increase the reader's comprehension:

Sheykhhasan M, Amini R, Soleimani Asl S, Saidijam M, Hashemi SM, Najafi R. Neuroprotective effects of coenzyme Q10-loaded exosomes obtained from adipose-derived stem cells in a rat model of Alzheimer's disease. Biomed Pharmacother. 2022 Aug;152:113224. doi: 10.1016/j.biopha.2022.113224. Epub 2022 Jun 6. PMID: 35679720.

b. Sheykhhasan M, Heidari F, Eslami Farsani M, Azimzadeh M, Kalhor N, Ababzadeh S, Seyedebrahimi R. Dual role of Exosome in neurodegenerative diseases: a review study. Curr Stem Cell Res Ther. 2023 Jul 26. doi: 10.2174/1574888X18666230726161035. Epub ahead of print. PMID: 37496136.

Response:

Thank you for your valuable comments. These references help us a lot, and we have added these sources to our revised manuscript. (see references 5-10)

5. The materials and Methods section requires more information: To confirm the success of ADSC isolation, in addition to the Flow cytometric analysis, it is necessary to confirm using the analysis of the trilineage differentiation potential of ADSC, including histochemical assays (oil red, alkaline phosphatase, and alcian blue stains). Why, there were no other staining for lipid, and chondrocytes? As a result, it is better to include the results of histochemical assays (alkaline phosphatase, and alcian blue stains) in the method and result sections of the present paper.

Response:

Thank you for your valuable comments. We followed the previous protocols to confirm the success of ADSC isolation, and based on the previous study, we believed that the ADSC could be identified by flow cytometric analysis. 

[Ismail HY, Shaker NA, Hussein S, Tohamy A, Fathi M, Rizk H, Wally YR. Cisplatin-induced azoospermia and testicular damage ameliorated by adipose-derived mesenchymal stem cells. Biol Res. 2023 Jan 19;56(1):2. doi: 10.1186/s40659-022-00410]

6. It is recommended that the authors describe the protocol for DiG-labeled ADSCs cells in method section.

Response:

Thank you for your valuable comments. The detailed protocol for Dil labeling has been added to our revised manuscript. “The procedures of Dil labeling were as follow: remove the supernatant from the culture dish and replace it with serum-free basal medium DMEM/F12 for cell cultivation. After 48 hours of incubation, collect the supernatant for extracellular vesicle extraction by the molecular department. For CM-DIL labeling of cells, digest the cells and suspend them in 10% DF/12 medium containing 5ul/ml of CM-DIL. Incubate the cell suspension at 37°C for 5 minutes and then place it at 4°C for 15 minutes to meet the experimental requirements. Provide a cell suspension with a concentration of 3×106 cells/ml labeled with CM-DIL for injection into the animal facility.”

7. In "Methods" section, why is the number of cells used in the cell therapy group 3x106, but in the Ex group, 100μg is used. Can the use of two different criteria be scientifically justified?

Response:

Thank you for your valuable comments. Due to the sizes of Ex, it is hard to describe the Ex by number. Therefore, we followed the previous study and using the above criteria. 

[Zhu LL, Huang X, Yu W, Chen H, Chen Y, Dai YT. Transplantation of adipose tissue-derived stem cell-derived exosomes ameliorates erectile function in diabetic rats. Andrologia. 2018 Mar;50(2). doi: 10.1111/and.12871.]

8. Figure 3 needs correction.

Response:

Thank you for your valuable comments. Figure 3 has been corrected in our revised manuscript.

9. In order to make the paper more interesting to read, I suggested that the authors could add one graphical abstract to the manuscript.

Response:

Response:

Thank you for your valuable comments. We tried to add a graphical abstract in our revised manuscript. 

Reviewer 2

1. kindly follow the rules of scientific writing and special attention for the references.

Response:

Thank you for your valuable comments. The whole manuscript has been revised with the guidance of native English speaker. 

2. kindly revise the manuscript for English errors and grammar.

Response:

Thank you for your valuable comments. The whole manuscript has been revised with the guidance of native English speaker. The whole manuscript has been revised with the guidance of native English speaker, and all the spelling mistakes have been corrected. 

3. Introduction: First page, last line (No. 21); please correct ADSC-EVs to (ADSC-EXs).

Response:

Thank you for your valuable comments. Corrected in our revised manuscript. 

4.Method:

4.1. Model establishment:T he authors did not mention the environment of animal housing, number and model of animals used in experiment. There are no references for the dose and rout of cisplatine and/or ADSC and/or ADSC-Ex administration. There are grammatical and spelling errors. How can you confirm testicular damage after 3 days of cisplatine administration?

Response:

Thank you for your wise advice. The description of environment of animal housing, number and model of animals used in experiment has been added to our revised manuscript. Moreover, the references for the dose and rout of cisplatine and/or ADSC and/or ADSC-Ex administration have been added to our revised manuscript. We selected 3 days as the time point of cisplatine administration following the pervious studies. 

[[Ismail HY, Shaker NA, Hussein S, Tohamy A, Fathi M, Rizk H, Wally YR. Cisplatin-induced azoospermia and testicular damage ameliorated by adipose-derived mesenchymal stem cells. Biol Res. 2023 Jan 19;56(1):2. doi: 10.1186/s40659-022-00410

Welsh JA, Van Der Pol E, Arkesteijn GJA, Bremer M, Brisson A, Coumans F, Dignat-George F, Duggan E, Ghiran I, Giebel B, Görgens A, Hendrix A, Lacroix R, Lannigan J, Libregts SFWM, Lozano-Andrés E, Morales-Kastresana A, Robert S, De Rond L, Tertel T, Tigges J, De Wever O, Yan X, Nieuwland R, Wauben MHM, Nolan JP, Jones JC. MIFlowCyt-EV: a framework for standardized reporting of extracellular vesicle flow cytometry experiments. J Extracell Vesicles. 2020 Feb 3;9(1):1713526. doi: 10.1080/20013078.2020.1713526

Makled MN, Said E. Tranilast abrogates cisplatin-induced testicular and epididymal injuries: An insight into its modulatory impact on apoptosis/proliferation. J Biochem Mol Toxicol. 2021 Aug;35(8):e22817. doi: 10.1002/jbt.22817

Ismail HY, Shaker NA, Hussein S, Tohamy A, Fathi M, Rizk H, Wally YR. Cisplatin-induced azoospermia and testicular damage ameliorated by adipose-derived mesenchymal stem cells. Biol Res. 2023 Jan 19;56(1):2. doi: 10.1186/s40659-022-00410-5. PMID: 36653814; PMCID: PMC9850593]

4.2. Cell identification How many rats were used for collecting adipose tissue? The authors did not mention any references for collection of ADSC form rats, is it a new technique established by the authors? How can you confirm the viability and count of ADSC after collection?

Response:

Thank you for your valuable comments. We used only one rats to obtain the adipose tissue to avoid bias caused by different individuals. We followed previous studies to collect ADSC from rats, and the count of ADSC has been identified by flow cytometric. 

[Taha MF, Hedayati V. Isolation, identification and multipotential differentiation of mouse adipose tissue-derived stem cells. Tissue Cell. 2010 Aug;42(4):211-6. doi: 10.1016/j.tice.2010.04.003. Epub 2010 May 21. PMID: 20483444.]

4.3. Exosome extraction: Again, the authors did not mention any references for collection of exosomes. The authors mentioned (Thaw the sample at 37°C.), but he did not mention which sample and what is the type of sample.

Response:

We are grateful for your valuable comments. The sample was the ADSC mentioned in 2.2 of the method part. We followed the protocols as follow:

[Welsh JA, Van Der Pol E, Arkesteijn GJA, Bremer M, Brisson A, Coumans F, Dignat-George F, Duggan E, Ghiran I, Giebel B, Görgens A, Hendrix A, Lacroix R, Lannigan J, Libregts SFWM, Lozano-Andrés E, Morales-Kastresana A, Robert S, De Rond L, Tertel T, Tigges J, De Wever O, Yan X, Nieuwland R, Wauben MHM, Nolan JP, Jones JC. MIFlowCyt-EV: a framework for standardized reporting of extracellular vesicle flow cytometry experiments. J Extracell Vesicles. 2020 Feb 3;9(1):1713526. doi: 10.1080/20013078.2020.1713526]

4.4. Flow Cytometry: Kindly please check the soundness of procedures and mention your guide for these procedures.

Response:

We are grateful for your valuable comments. Checked, thank you.

4.5. Evaluation of ADCS-Ex: No references mentioned for the technique of ADCS-Ex evaluation.

Response:

We are grateful for your valuable comments. Added to our revised manuscript.

[Welsh JA, Van Der Pol E, Arkesteijn GJA, Bremer M, Brisson A, Coumans F, Dignat-George F, Duggan E, Ghiran I, Giebel B, Görgens A, Hendrix A, Lacroix R, Lannigan J, Libregts SFWM, Lozano-Andrés E, Morales-Kastresana A, Robert S, De Rond L, Tertel T, Tigges J, De Wever O, Yan X, Nieuwland R, Wauben MHM, Nolan JP, Jones JC. MIFlowCyt-EV: a framework for standardized reporting of extracellular vesicle flow cytometry experiments. J Extracell Vesicles. 2020 Feb 3;9(1):1713526. doi: 10.1080/20013078.2020.1713526]

4.6. Immunohistochemistry: What is the method of tissue preservation for immunohistochemistry?

The word (100% ethanol) is repeated twice, please check and correct with referral to reference of this technique. Please mention the source of chemicals and kits used in this study.

Response:

We are grateful for your valuable comments. Wrong edit here. Revised in our revised manuscript. 

4.7. ELISA detection: kindly check references and grammatical errors. The authors did not declare which type of ELISA used and what they searched for?

Response:

We are grateful for your valuable comments. Added to our revised manuscript, and the errors have been checked and corrected. 

[ELISA kits were used to identify the levels of follicle-stimulating hormone (FSH, E-EL-RO391c，abscience) and luteinizing hormone (LH, E-EL-RO026c，Elabscience), testosterone (T, E-OSEL-R0003，ascience), malondialdehyde (MDA, E-EL-0060c,Elabscience).]

5. Statistical analysis: What is the post hoc. test was used in your study?

Response:

We are grateful for your valuable comments. We used B-H method to adjusted our p-value, and we forget to mention it in our method part. This part has been added to our revised manuscript. 

6. Discussion: There is a similar study and will be helpful in your manuscript and you did not mention it in your study: (Cisplatin-induced azoospermia and testicular damage ameliorated by adipose-derived mesenchymal stem cells) Biol Res. 2023 Jan 19;56(1):2. doi: 10.1186/s40659-022-00410-5.

Response:

We are grateful for your valuable comments. More discussion about this study has been added to our revised manuscript.

Reviewer 3

1. Major revision requirieds.

Response:

Thank you for your valuable feedback. We have carefully considered the comments and made substantial revisions to address the concerns raised by the reviewers.

Reviewer 4

1. What is the difference between this experiment and previous one (Like the article below)? Mention in discussion.

Meligy FY, Abo Elgheed AT, Alghareeb SM. Therapeutic effect of adipose-derived mesenchymal stem cells on Cisplatin induced testicular damage in adult male albino rat. Ultrastruct Pathol. 2019;43(1):28-55. doi:10.1080/01913123.2019.1572256. Epub 2019 Feb 9. PMID: 30741078.

Response:

Thank you for your valuable feedback. More information on this has been added to our revised manuscript. 

2. The last paragraph of introduction the author writes that “However, few studies reported the effect of ADSC-Ex treatment on testicular injury. In this study, we aimed to extract the ADSC-Ex and explore the impact of ADSC-Ex on rat models with testicular injury.” Ok. Please refer to those articles and what has not been done before that you focused on in this study.

Response:

Thank you. Added accordingly. 

[Meligy FY, Abo Elgheed AT, Alghareeb SM. Therapeutic effect of adipose-derived mesenchymal stem cells on Cisplatin induced testicular damage in adult male albino rat. Ultrastruct Pathol. 2019;43(1):28-55.]

3. Provide a reference for the testicular destruction model by cisplatin solution.

Response:

Thank you. Added accordingly. 

[Ismail HY, Shaker NA, Hussein S, Tohamy A, Fathi M, Rizk H, Wally YR. Cisplatin-induced azoospermia and testicular damage ameliorated by adipose-derived mesenchymal stem cells. Biol Res. 2023 Jan 19;56(1):2. doi: 10.1186/s40659-022-00410]

4. what is the volume of injection of cisplatin and ADSCs cells?

Response:

Thank you. The volume was determined by the mouse’s weight, as we described in 2,1, Methods part. 

5. On what basis were these concentrations of 3*10^6 CM-DiI-labeled ADSCs cells and 100μg of ADSC-Ex used for injection? What is CM-DiI?

Response:

Thank you. The reference of the concentrations has been added to our manuscript. The information on CM-Dil has been added to our manuscript.

[The procedures of Dil labeling were as follow: remove the supernatant from the culture dish and replace it with serum-free basal medium DMEM/F12 for cell cultivation. After 48 hours of incubation, collect the supernatant for extracellular vesicle extraction by the molecular department. For CM-DIL labeling of cells, digest the cells and suspend them in 10% DF/12 medium containing 5ul/ml of CM-DIL. Incubate the cell suspension at 37°C for 5 minutes and then place it at 4°C for 15 minutes to meet the experimental requirements. Provide a cell suspension with a concentration of 3×106 cells/ml labeled with CM-DIL for injection into the animal facility] 

6. What sedation, anesthesia or anesthesia protocol was used to remove subcutaneous white adipose tissue?

Response:

Thank you. More information on anesthesia has been added to our revised manuscript. [Six-week-old SD rat was disinfected with 75% alcohol for 3 minutes.]

7. What culture mediums were used? Mention the names of the companies related to the preparation of materials? Why didn't you use penicillin/streptomycin?

Response:

Thank you. Wrong edit here. Information on culture mediums and antibiotics has been added to our revised manuscript. [The digestion was stopped by adding a complete containing 10% fetal bovine serum (FBS) and 1% penicillin–streptomycin (P/S). The culture medium, FBS, P/S were purchased from Gibco (ThermoFisher, Shanghai, China).]

8. The order of method steps in the method is not clear. First Model establishmen then Cell identification and evaluation and then Immunohistochemistry and Immunohistochemistry?!!!! ELISA detection for which protein? From which company? Also, the order of the result should be based on the order of the steps of the work method.

Response:

Thank you. Added to our revised manuscript. 

[ELISA kits were used to identify the levels of follicle-stimulating hormone (FSH, E-EL-RO391c，abscience) and luteinizing hormone (LH, E-EL-RO026c，Elabscience), testosterone (T, E-OSEL-R0003，ascience), malondialdehyde (MDA, E-EL-0060c,Elabscience).]

9. What is NTA? When first expressed, it should not be in the form of abbereviation.

Response: 

Thank you for your valuable feedback. NTA was “Nanoparticle Tracking Analysis”. Added to our revised manuscript. 

10. In order to confirm the stem cells, Trilineage Differentiation should be done, but in this study, only differentiation to osteogenesis was done.

What was the difference between the results of using ADSCs cells and of ADSC-Ex? Which one is better?

You don’t use Post hoc to analyze the results of your experimental data?

Response:

We are grateful for your valuable comments. We used B-H method to adjusted our p-value, and we forget to mention it in our method part. This part has been added to our revised manuscript.

---

## [Decision Letter · Decision Letter 1]

13 Dec 2023

PONE-D-23-25567R1Roles of adipose-derived stem cells and derived exosomes in therapeutic applications to testicular injury caused by cisplatinPLOS ONE

Dear Dr. wu,

Thank you for submitting your manuscript to PLOS ONE. After careful consideration, we feel that it has merit but does not fully meet PLOS ONE’s publication criteria as it currently stands. Therefore, we invite you to submit a revised version of the manuscript that addresses the points raised during the review process. Please submit your revised manuscript by Jan 27 2024 11:59PM. If you will need more time than this to complete your revisions, please reply to this message or contact the journal office at plosone@plos.org. Please include the following items when submitting your revised manuscript:A rebuttal letter that responds to each point raised by the academic editor and reviewer(s). You should upload this letter as a separate file labeled 'Response to Reviewers'.A marked-up copy of your manuscript that highlights changes made to the original version. You should upload this as a separate file labeled 'Revised Manuscript with Track Changes'.An unmarked version of your revised paper without tracked changes. You should upload this as a separate file labeled 'Manuscript'.

We look forward to receiving your revised manuscript.

Kind regards,

Ahmed E. Abdel Moneim

Academic Editor

PLOS ONE

Journal Requirements:

Additional Staff Editor Comments:

Please include methods of animal sacrifice in the Methods section.

Reviewers' comments:

Reviewer's Responses to Questions

**Comments to the Author**

1. If the authors have adequately addressed your comments raised in a previous round of review and you feel that this manuscript is now acceptable for publication, you may indicate that here to bypass the “Comments to the Author” section, enter your conflict of interest statement in the “Confidential to Editor” section, and submit your "Accept" recommendation.

Reviewer #1: All comments have been addressed

Reviewer #2: All comments have been addressed

Reviewer #3: (No Response)

2. Is the manuscript technically sound, and do the data support the conclusions?

Reviewer #1: Yes

Reviewer #2: Yes

Reviewer #3: Yes

3. Has the statistical analysis been performed appropriately and rigorously? 

Reviewer #1: Yes

Reviewer #2: Yes

Reviewer #3: Yes

4. Have the authors made all data underlying the findings in their manuscript fully available?

Reviewer #1: Yes

Reviewer #2: Yes

Reviewer #3: Yes

5. Is the manuscript presented in an intelligible fashion and written in standard English?

Reviewer #1: Yes

Reviewer #2: Yes

Reviewer #3: Yes

6. Review Comments to the Author

Reviewer #1: The authors completely responded to the observations made previously. They also inserted useful and clarifying information. Thank you for editing the manuscript. Now, the manuscript is much better than the former.

Reviewer #2: The authors have addressed all comments. the manuscript is technically sound, and it may be accepted for publication.

Reviewer #3: Minor revision

1. Abbreviations should be explained what they stand for the first time they are used. It is generally not accepted to use

abbreviations in abstracts unless they are spelled in full at least once.

2.Which one? SD (Sparaque Dawley) rats or SD (Standard Deviation)

3. Which one? rats or mice.

4. Please add study limitations

7. PLOS authors have the option to publish the peer review history of their article (what does this mean?). If published, this will include your full peer review and any attached files.

Reviewer #1: **Yes: **Mohsen Sheykhhasan

Reviewer #2: No

Reviewer #3: No

---

## [Author Response · Author response to Decision Letter 1]

19 Dec 2023

1. Abbreviations should be explained what they stand for the first time they are used. It is generally not accepted to use abbreviations in abstracts unless they are spelled in full at least once.

Response:

Thank you for your feedback. I have addressed this concern by ensuring that all abbreviations are spelled out in full the first time they are used in the abstract. This will provide clarity for the readers and adhere to the accepted standards for abstracts.

2.Which one? SD (Sparaque Dawley) rats or SD (Standard Deviation).

Response:

Thank you for the clarification. I have ensured that all instances of "SD" are consistently used to represent "Standard Deviation" and that the full term "Sprague Dawley" is used when referring to the rats. This will help to avoid any confusion regarding the use of the abbreviation. Thank you for your input.

3. Which one? rats or mice.

Response:

Thank you for bringing this to my attention. Sprague Dawley rats were used as experimental animals. I have reviewed and corrected the error in the manuscript, ensuring that the correct terms "Sprague Dawley rats" or “rats” were used consistently throughout the entire study. I appreciate your attention to detail and apologize for any confusion caused by the initial error. Thank you for your feedback.

4. Please add study limitations.

Response:

Thank you for your comments. We have revised the manuscript to include a section on study limitations, addressing the points you raised.

This part has been added:

[Despite the promising results of our study, there are several limitations that should be considered. Firstly, the current study was performed in a rat model of cisplatin-induced testicular injury, and further studies are needed to confirm the therapeutic effects of adipose-derived stem cells (ADSCs) and their exosomes in humans. Secondly, the mechanisms underlying the therapeutic effects of ADSCs and their exosomes on testicular injury are still not fully understood, and more research is needed to elucidate these mechanisms. Thirdly, the long-term safety and potential side effects of ADSCs and their exosomes need to be carefully evaluated in future studies. Finally, the cost and practicality of using ADSCs and their exosomes as a therapeutic approach for testicular injury in clinical settings need to be further explored. Moreover, several challenges need to be addressed before ADSC-Ex can be used in clinical setting]

---

## [Decision Letter · Decision Letter 2]

27 Dec 2023

Roles of adipose-derived stem cells and derived exosomes in therapeutic applications to testicular injury caused by cisplatin

PONE-D-23-25567R2

Dear Dr. wu,

We’re pleased to inform you that your manuscript has been judged scientifically suitable for publication and will be formally accepted for publication once it meets all outstanding technical requirements.

Kind regards,

Ahmed E. Abdel Moneim

Academic Editor

PLOS ONE

Additional Editor Comments (optional):

Reviewers' comments:

Reviewer's Responses to Questions

**Comments to the Author**

1. If the authors have adequately addressed your comments raised in a previous round of review and you feel that this manuscript is now acceptable for publication, you may indicate that here to bypass the “Comments to the Author” section, enter your conflict of interest statement in the “Confidential to Editor” section, and submit your "Accept" recommendation.

Reviewer #3: (No Response)

2. Is the manuscript technically sound, and do the data support the conclusions?

Reviewer #3: Yes

3. Has the statistical analysis been performed appropriately and rigorously? 

Reviewer #3: Yes

4. Have the authors made all data underlying the findings in their manuscript fully available?

Reviewer #3: Yes

5. Is the manuscript presented in an intelligible fashion and written in standard English?

Reviewer #3: Yes

6. Review Comments to the Author

Reviewer #3: (No Response)

7. PLOS authors have the option to publish the peer review history of their article (what does this mean?). If published, this will include your full peer review and any attached files.

Reviewer #3: No

---

## [Editor Report · Acceptance letter]

26 Jan 2024

PONE-D-23-25567R2 

PLOS ONE

Dear Dr. wu, 

I'm pleased to inform you that your manuscript has been deemed suitable for publication in PLOS ONE. Congratulations! Your manuscript is now being handed over to our production team.

Kind regards, 

on behalf of

Dr. Ahmed E. Abdel Moneim 

Academic Editor

PLOS ONE